# Towards an Evidence-Based Classification System for Para Dressage: Associations between Impairment and Performance Measures

**DOI:** 10.3390/ani13172785

**Published:** 2023-08-31

**Authors:** Sarah Jane Hobbs, Jill Alexander, Celeste Wilkins, Lindsay St. George, Kathryn Nankervis, Jonathan Sinclair, Gemma Penhorwood, Jane Williams, Hilary M. Clayton

**Affiliations:** 1Research Centre for Applied Sport, Physical Activity and Performance, University of Central Lancashire, Preston PR1 2HE, UK; jalexander3@uclan.ac.uk (J.A.); lbstgeorge@uclan.ac.uk (L.S.G.); jksinclair@uclan.ac.uk (J.S.); 2Sport and Exercise Department, Hartpury University, Hartpury, Gloucester GL19 3BE, UK; celeste.wilkins@hartpury.ac.uk; 3Equine Department, Hartpury University, Hartpury, Gloucester GL19 3BE, UK; kathryn.nankervis@hartpury.ac.uk (K.N.); jane.williams@hartpury.ac.uk (J.W.); claytonh@msu.edu (H.M.C.); 4Department of Animal and Agriculture, Hartpury University, Hartpury, Gloucester GL19 3BE, UK; gemma.penhorwood2@hartpury.ac.uk

**Keywords:** Para, dressage, impairment, classification, athlete, horse, assessment, trunk, stability, asymmetry

## Abstract

**Simple Summary:**

In the sport of Para dressage, the combined performance of the Para athlete and the horse is judged subjectively. To provide a level playing field for athletes with a wide range of impairments, Para dressage athletes are classified based on the degree to which their impairment impacts sports performance. This study aimed to evaluate the effects of impairment on objective performance measures in Para athletes as a step towards developing an evidence-based classification system. Two groups of athletes comprising twenty-one elite Para athletes (classified grades I to V) and eleven non-disabled athletes (competing at Prix St. Georges or Grand Prix) were measured while performing a 2 min custom dressage test on a riding simulator and clinically assessed using a battery of impairment assessment tools. Statistical analysis considered the extent to which performance could be predicted from impairment measures. Impairment assessment tools related to sitting function could predict the impact of impairment on performance in Para athletes but not in non-disabled athletes. These findings provide evidence of sport-specific impairment assessments that could be used to enhance athlete classification. These findings pave the way for further studies using similar approaches to enhance the objectivity of classification between grades.

**Abstract:**

This study follows a previously defined framework to investigate the impact of impairment on performance in Para dressage athletes. Twenty-one elite Para dressage athletes (grades I to V) and eleven non-disabled dressage athletes (competing at Prix St. Georges or Grand Prix) participated. Data were collected in two phases: performing a two minute custom dressage test on a riding simulator while kinematic data were synchronously collected using inertial measurement units (2000 Hz) and optical motion capture (100 Hz), and clinically assessed using a battery of impairment assessment tools administered by qualified therapists. Impairment and performance measures were compared between Para and non-disabled athletes. Significant differences between athlete groups were found for all impairment measures and two performance measures: simulator trunk harmonics (*p* = 0.027) and athlete trunk dynamic symmetry (*p* < 0.001). Impairment assessments of sitting function and muscle tone could predict 19 to 35% of the impact of impairment on performance in Para athletes but not in non-disabled athletes. These findings provide the basis for a robust, scientific evidence base, which can be used to aid in the refinement of the current classification system for Para dressage, to ensure that it is in line with the International Paralympic Committee’s mandate for evidence-based systems of classification.

## 1. Introduction

Para dressage is the only equestrian discipline represented in the Paralympic Games. The athlete partners with the horse to complete a floorplan of movements (the dressage ‘test’) in an arena. Like other Paralympic sports, the athlete undergoes a classification process to determine their eligibility to compete and to assign a grade that groups athletes based on the impact of their impairment on sports performance. This process aims to ensure, as much as possible, that sporting excellence determines the outcome of competition [1,2,3,4]. The Fédération Equestre Internationale (FEI) is the governing body for Para dressage and is responsible for the development and regulation of its sport-specific classification system. Since its inception, the classification system has been based on the Meaden Profile System [5]. In this system, athletes are classified into five grades determined using the results of a series of seated clinical functional tests as well as ridden observations conducted by qualified classifiers [4,6]. Although classification aims to ensure fair competition by grouping athletes with impairments that result in similar activity limitations, equitable classification poses a key challenge in Para dressage, primarily due to the unique influence of the horse on the outcomes of sports performance [7,8].

Given the importance of the classification system within Para sport, the International Paralympic Committee (IPC) published the Athlete Classification Code [1], which mandates the development of evidence-based classification systems across all Paralympic sports. In recognition of this, [2] proposed a framework of actions for research programs that aim to develop evidence-based, sport-specific classification systems. The actions include (a) identifying sport-specific determinants of performance, (b) developing robust measures of impairment alongside standardized measures of key performance determinants, (c) assessing the relative strength of association between measures of impairment and performance, and (d) using these outcomes to determine the minimum impairment criteria and sports class profiles [9,10].

The current Para dressage classification system focuses on the assessment of impairment, but, as with many other Paralympic sports, the system requires greater emphasis on objective measures that quantify the extent of activity limitation due to the impairment(s) [11]. Thus, in 2018, the FEI funded a research project designed to follow the framework set out by [9] as a first step in providing a strong, scientific evidence-base for the Para dressage classification system [11]. This project has already produced a scoping review of published studies that investigated key determinants of dressage performance [7], a study of stakeholder perceptions regarding the key determinants of and impact of impairment on Para dressage performance [8], and a research synthesis of existing clinical impairment assessment tools that are relevant to eligible impairment types, performance measures, and activity limitations for Para dressage [12].

The synthesis [12] recommended the following robust impairment assessment tools to take forward into further studies: the Function in Sitting Test (FIST) [13], the Trunk Impairment Scale (TIS) [14], the Scale of Assessment and Rating of Ataxia (SARA) [15], the modified Ashworth Scale, or “Re-modified Ashworth Scale” (R-MAS) [16], and handheld dynamometry (HHD) measures [17]. These tools had acceptable psychometric properties, captured the physical requirements for dressage performance, and were considered to meet the practical demands of athlete classification [12]. The FIST is considered to be a reliable measure of activity and performance-based deficits in sitting function [13], while the TIS is considered a reliable measure of trunk control in sitting [14]. SARA, versions of the Ashworth and modified Ashworth Scale, and HHD measures are used in other Para sport classification systems [18,19,20,21], measuring aspects of coordination, muscle tone, and strength, respectively [12]. The tools also provide access to substantial data evaluating their reliability and repeatability across a range of conditions that are present within the Para dressage population, adding to the strength of their use within this study as impairment assessment measures.

Identifying the key determinants of performance and understanding the impact of different impairments on performance is not an easy task for Para dressage, given the nature of the sport and the range of eligible impairments [6]. In these scenarios, qualitative methods have been recommended to obtain athlete and stakeholder input for refining key determinants of performance [9]. As such, Para dressage stakeholders (athletes, classifiers, coaches, and judges) were interviewed, and these interactions highlighted the key role of the horse in determining the outcomes of Para dressage performance [8]. This is not surprising, as the scoring system for a dressage test is based on subjective scoring by judges that is guided by specific directives that relate to the performance of the horse-athlete dyad. Thus, the overall dressage score is influenced by many factors related to both equine and human athletes. The variability between horses’ inherent conformation and behavior can contribute to gait kinematic differences and can influence the score awarded, for example, stride length [22]. The accuracy with which the athlete instructs the horse to perform the test also influences the score, although stakeholders in Para dressage perceive the inherent quality of the horse’s gaits as highly influential in gaining a higher score [8]. Several quantitative measures of horse performance in different gaits and movements were identified that can be influenced, for better or worse, by the skill of the athlete [7]. These were later refined into determinants of dressage performance: harmonics, stability, coordination, and symmetry, and tested using dressage horses and athletes [23,24] to inform the design of this study.

Following on from the previous studies that form the FEI Classification Research project [7,8,12], and additional studies to identify suitable quantitative measures of performance [23,24,25], this article presents findings from the final study of the overarching research project. It is informed by the reviews [7,12], semi-structured interviews [8], and original studies [23,24,25] that have preceded it, extensive consultation and input from classifiers and physiotherapists, and pilot testing. This cumulative study aims to investigate and define measures of impairment, measures of performance, and the strength of association between impairment and activity limitation in Para dressage and non-disabled athletes, as described by [2]. In accordance with [9], performance and impairment measurements from non-disabled athletes were used to develop a normative dataset for comparison with Para athletes. The objectives were (1) to compare the outcome of a clinical battery of recommended impairment assessment tools for Para dressage between Para and non-disabled athletes [12]; (2) to compare non-disabled and Para athlete performance using a custom dressage test on a riding simulator; and (3) to objectively determine if, how, and to what extent impairment severity measured with these tools influences simulated riding performance.

It was hypothesised that (1) the impact of impairment on maintaining coordinated, symmetrical stride-to-stride motion of the trunk and pelvis will be measurable between Para and non-disabled athletes with comparable skill levels [26,27,28]; (2) coordination during simulated riding will be independent of muscle strength; and (3) performance will be impacted more in athletes with activity limitations that affect the trunk and pelvis, measurable using clinically validated tests of sitting function.

## 2. Materials and Methods

Ethical approval for the study was obtained through the University of Central Lancashire’s Ethics Committee (approval reference: STEMH 910 Phase 2). Informed consent was obtained from all participants prior to the study.

### 2.1. Participants

Due to the complexity of impairment/activity limitation of Para athletes, a diverse sample of para dressage athletes was recruited for the study. A total of twenty-one classified, elite Para dressage athletes representing Grades I to V participated in the study (Table 1). Para dressage athletes had a minimum of 5 years’ experience competing at international (CPEDI) level competition and had a Confirmed (C), Review (R) or Review with Fixed Date (RFD) FEI classification status. Eleven experienced, non-disabled dressage athletes also participated in the study (Table 1). The non-disabled group of athletes were competing at the Prix St. Georges (PSG) or higher at the time of the study. This criterion was necessary to reduce the confounding effects of skill, as training for elite performance is known to enhance motor control and physiological function [29], and changes in the athletes’ riding posture can only really be detected at an advanced level [30].

Para athletes in this study who volunteered came from Great Britain Ireland, and United States of America. Of the athletes from these countries that currently have registered FEI classification status, forty-four met the eligibility criteria. The Para athlete sample in this study therefore represented 48% of the population from these three countries and approximately 12% of the eligible Para dressage athlete population worldwide. Non-disabled athletes represented a small reference sample of high-level dressage athletes from Great Britain, Ireland and the United States of America.

### 2.2. Data Collection

Data were collected in two phases: a performance assessment phase (on a riding simulator) followed by a clinical assessment phase, collected on the same day (with a break between phases, where required) for all Para and non-disabled athletes except for one Para athlete. For this Para athlete, the severity of their impairment necessitated that clinical and performance assessment phases be undertaken on separate days to avoid fatigue. Data were collected over 26 days at Hartpury University (Gloucester, UK) between October 2021 and July 2022.

#### 2.2.1. Performance Assessment

Kinematic data were collected from participants during simulated riding using an Eventing Simulator (Racewood Ltd., Tarporley, UK), which can simulate equine gaits and movements that are performed during advanced dressage. It is equipped with a screen that provides a virtual 20 × 40 m dressage arena with standard arena letter markers for practicing tests (Figure 1 and Figure 2). During testing, participants were required to wear a riding helmet that met current British Standards Institute safety standards (PAS 015: (1998/2011) with the BSI kitemark). Depending on the Para athlete’s needs and to ensure safety, compensatory aids were permitted if the athlete required them to undertake the testing. All Para and non-disabled athletes except one Para athlete rode in the same dressage saddle (Albion SLK, Albion, Walsall, UK), which was secured to the riding simulator and checked between each rider for fit and balance in a standardized manner. The compensating aids that were permitted during testing included: looped reins (n = 6), seat saver (n = 3), no stirrups (n = 3), enclosed stirrups (n = 1), stirrup iron to girth straps (n = 1), whips (n = 6), hand hold (soft and hard) (n = 2), and own saddle (n = 1). Seven Para athletes used no compensating aids; five Para athletes that used whips also completed the test without whips; one Para athlete needed the rein tension to be more sensitive; and five Para athletes had a caller.

All Para and non-disabled athletes were permitted to observe the motion of the simulator prior to mounting and asked to wear a tightly fitting tunic to which the trunk sensors could be attached. All Para and non-disabled athletes mounted the stationary simulator using a large mounting platform with assistance from the researchers and/or the Para athlete’s accompanying friend/relative/colleague where necessary, and under the instruction and discretion of the participant. Once mounted, each athlete underwent a warm-up period to allow acclimatization to the simulator’s motion, the virtual reality 20 × 40 arena, and the sensitivity of the rein aids. The speed of the simulator (walk, trot, and halt) was controlled by one of two researchers throughout the study (K.N., C.M.), and trotting was optional for grade I Para athletes, as trotting is not included in a grade I dressage test. A maximum of 10 min was permitted for warm-up/acclimatization.

Following the warm-up, retro-reflective markers and inertial measurement unit (IMU) sensors (Delsys Inc., Natick, MA, USA) were positioned over anatomical landmarks while the athlete remained mounted on the simulator. To collect kinematic data from the trunk, pelvis, and head segments of each athlete, retro-reflective markers were applied bilaterally to the following anatomical landmarks: acromion process, iliac crest, anterior superior iliac spine (ASIS), posterior superior iliac spine (PSIS), between left and right PSIS, and a cluster of 4 markers fixed to the tunic at approximately T3–T5 level. IMU sensors were applied to the sacrum, trunk (tunic at approximately T3–T5 level), and head (back of the helmet using an elasticated band). To measure the motion of the simulator, retroreflective markers were placed on the poll and croup and bilaterally on the shoulders, hips, and head. One IMU sensor was applied to the simulator trunk behind the saddle. Markers and IMU sensors were attached to each anatomical location using hypoallergenic tape and flexifoam straps, except for the trunk cluster, which was attached to the rider’s tunic with velcro.

Three-dimensional (3D) optical motion capture (100 Hz) and IMU (2000 Hz) data were synchronously captured in Qualisys Track Manager (QTM, Qualisys AB, Göteborg, Sweden) using an external trigger (Delsys Trigger Module, Delsys Inc., Natick, MA, USA) during a static trial and during the custom dressage test. The static trial was initially collected with the participants sitting still in their normal riding position on the stationary simulator. The athlete was then provided with the dressage test and instructed to “ride for scores of 10”, which is the highest score that is awarded in dressage, within the simulated 20 × 40 m dressage arena. All Para and non-disabled athletes were required to steer the simulator within the virtual arena during specified changes in direction (turning left and right). The dressage test is outlined in Figure 2 and includes walk, halt-walk, and walk-halt transitions, changing direction (turning left and right), and trot, halt-trot, and trot-halt transitions. Walk was principally used in the test, as it has previously been described as more difficult to ride than trot or canter for the athlete, based on a lack of coordination and larger phase differences between the trunk motion of the athlete and the trunk motion of the horse [31]. Kinematic data were collected for the duration of the dressage test, and events were triggered at locations/arena markers C, B, E, and A to ensure that the data could be linked to the dressage performance. The simulator testing protocol, including mounting, warm up/acclimatization, marking up, data collection, removal of markers, and dismounting, lasted approximately 15 min. The dressage test was approximately 2 min long. The testing protocol was designed with the duration of a typical Para dressage test in mind, where the durations of grade I to III and grade IV to V tests are 4.5 and 5 min, respectively [32].

#### 2.2.2. Impairment Assessments

Following the performance assessment, all Para and non-disabled athletes underwent a battery of impairment assessment measures. The suite of impairment assessment measures was initially identified by [12] and developed into a training manual [33]. The manual was refined through extensive consultation with fully accredited and highly experienced FEI classifiers and physiotherapists. These exercises highlighted the following clinical impairment assessment measures to be best suited for measuring eligible impairments in Para dressage athletes [12]: R-MAS [16], FIST [13], SARA [15], TIS [14], and HHD [17]. In accordance with the current FEI Para dressage classification system [4,6], all participants were assessed by one of three qualified and accredited Physiotherapists/Sports Therapists (J.A., L.N., and A.H.) who were trained to undertake the testing.

It is important to note that the clinician performing the impairment assessments was provided with the current grade of the athlete but was unaware of any diagnosed impairment/medical history that may have introduced bias during this part of the testing.

Impairment assessments were performed in the same private room at Hartpury University on a hydraulic plinth, with the order of impairment assessments randomized. To reduce the number of times the athletes had to change position on the plinth and mitigate any physical and/or mental fatigue, the seated impairment assessments were always performed first. This was followed by any assessments that required a supine position on the plinth. In this format, the order of FIST, TIS, and SARA (except for the ‘shin to heel slide’ component) were conducted first and in a randomized order, followed by seated assessment of trunk HHD, elbow R-MAS and HHD, shoulder R-MAS and HHD, then hip R-MAS and HHD (external rotation in sitting and then adduction in supine). Finally, the SARA “shin to heel slide” component was performed in a supine position. Scores for each test were input directly into an electronic scoring sheet in Excel (Microsoft Corporation, Redmond, WA, USA) developed specifically for this study to include scores for all components utilized for each impairment assessment [33]. Each impairment assessment has an independent scoring key to grade the performance, and most assessments (FIST, TIS, SARA, and R-MAS) provided a score that included the option of ‘unable to perform’. A score of zero was given for average and peak HHD metrics when athletes were unable to perform these assessments. This was performed to mitigate missing data for the final data analysis. Each impairment measure was performed according to published instructions [33]. Standardization and practice of the impairment assessments were discussed, performed, and evaluated between the therapists prior to and between testing days to ensure consistency throughout the testing phases.

### 2.3. Data Analysis

#### 2.3.1. Data Analysis for Performance Measures

Following testing, anonymized data were tracked in QTM software (version 2021.1, Qualisys AB, Göteborg, Sweden) and imported to Visual3D (version 6.03.0, C-motion Inc., Germantown, MD, USA) software for analysis. Marker trajectories were smoothed with a low-pass, 4th-order zero-lag Butterworth filter [34]. Trunk and pelvis segments were modelled in six degrees of freedom (they were allowed to move freely in space through three orthogonal rotations and three orthogonal translations) using the Calibrated Anatomical Systems Technique [35]. The static trial was used to create a model for each athlete and for the simulator. The athletes’ trunk segment was constructed using the acromion process and iliac crest markers. Upper trunk tracking markers were used (acromion processes and tracking cluster markers) to track the trunk segment. The CODA pelvis model (Charnwood Dynamics Ltd., Leicestershire, UK) was used to construct the pelvis segment. Pelvis center of mass (COM) position in the anterior–posterior direction was modelled by 50% of the distance between ASIS and PSIS markers. The pelvis was tracked using ASIS, PSIS, iliac crest, and sacrum markers. The trunk of the simulator was defined using the simulator shoulder and hip markers, and together with the marker on the croup, these tracked the simulator trunk segment. Additional tracking markers on the simulator head were used to assist in identifying turning during the dressage test.

The rotation between the simulator segment and the lab coordinate system and between the trunk and pelvis segments of the athlete was calculated using an XYZ cardan sequence. For each segment, the Z axis defined the long axis of the segment; accordingly, X was pitch or flexion-extension rotation, Y was lateral bending (trunk and pelvis) and yaw (simulator trunk), and Z was axial rotation (trunk and pelvis) and roll (simulator trunk). The magnitude of the three-dimensional (3D) rotational motion of the simulator within the lab environment and the 3D rotation between the trunk and pelvis segments were calculated as described by [24]. The resultant acceleration was calculated from the IMU data for each sensor separately using the same technique [24]. The resultant acceleration signals were then centered around zero by removing the mean offset from the first quiet period during a halt.

Stride segmentation was performed using maximal vertical displacement of the croup marker. As two peaks occur over each stride, the chosen peak corresponds to the point where mediolateral displacement to the right is ascending towards a maximum. Three non-consecutive strides of walk, three right turns (at arena marker C, in the arena corner between C and M, and at arena marker B), three left turns (at arena marker E, in the arena corner between K and A, and at arena marker A), and three non-consecutive straight trot strides (between A and C during steady state motion) were extracted. In addition, four halts, defined as the last stride before the halt at a point where <2 mm of consistent vertical displacement was exhibited, were extracted. There were some exceptions where consistent vertical displacement was not observed but instead drifted due to changes in athlete posture over the halt duration. In these cases, consistent displacement allowing for drift was estimated.

Performance measurements for the athlete and simulator were based on the measurements defined in [23,24]. To assess “Harmonics”, the signal power of the frequency distribution (SPower_sf_) and harmonic ratio (SRatio_sf_) of the 3D rotation of the trunk of the simulator were calculated for walk and trot strides at the simulator stride frequency using a base frequency of 0.68 Hz and 0.87 Hz (corresponding to slower motion cycles than stride frequencies reported for dressage horses, 0.88 Hz for walk and 1.22 Hz for trot [36]). Actuators move the simulator’s trunk at regular frequencies in vertical, cranio-caudal, and medio–lateral directions [25], providing a regular and repeated pattern of rotation for each simulated stride. Changes to the simulator’s motion pattern can be influenced by the athlete and are detectable from the frequency content of the signal [31,37], specifically the amount of signal power or the ratio between the even (sine) and odd (cosine) signal components at the stride frequency, as previously described by [24]. If the athlete restricts the motion, indicating a difference in coordinated motion between athlete and simulator, the signal power decreases and the ratio between signal components deviates from expected values of 1.2 to 1.3 for walk (unpublished data).

“Stability” was evaluated by calculating the root mean square (RMS) resultant acceleration [38] of the simulator trunk and athlete’s head during each walk-halt transition. The average RMS acceleration for the four halts was determined for each segment, and the percentage difference in the average RMS acceleration of the head compared to the simulator was calculated. Positive values indicated that the head accelerated less than the simulator, and negative values indicated that the head accelerated more than the simulator.

“Within-athlete coordination variability” was assessed by evaluating the detrended angular impulse of 3D trunk to pelvis rotation of the nine strides of straight-line walk, left and right turns together, and the three strides of trot separately. To compare between athletes, the average absolute deviation [36] over the nine strides for walk and three strides for trot were calculated.

“Symmetry” was evaluated using the average COM motion over the nine walk strides of the pelvis and trunk in the horizontal plane, relative to the pelvis static position. The midpoint of the dynamic pattern was ascertained by identifying the crossing point between the left and right patterns. Following this, the right pattern was folded over the left, and the average of the two patterns at each timepoint was calculated based on [39]. “Dynamic symmetry” was then represented by the sum of the square root between the squared difference of the average pattern for the left and right sides. As the data were plotted relative to the static pelvic COM position, a resultant vector was also calculated from the static pelvic position to the mid-point of the dynamic COM pattern, providing a “Symmetry vector”. These data were then normalized to pelvic width for the pelvis and trunk length for the trunk. See Figure 3 for further details.

#### 2.3.2. Data Analysis for Impairment Measures

Individual elements and total scores from the impairment assessment testing were collated in Excel (Microsoft Corporation, Redmond, WA, USA) in preparation for statistical analysis.

Existing classification information from each athlete was used to assign athletes to appropriate impairment sub-groups (Table 1).

### 2.4. Statistical Analysis

Descriptive statistics of ensemble averages for all measurements (mean, standard deviation, and coefficient of variation (%COV)) were calculated for Para and non-disabled athletes. These were also separated by grade for Para athletes and by level for non-disabled athletes.

To fulfil objectives (1) and (2), between subjects, General Linear Models (GLM) with simple bootstrapping due to the non-normal distribution of data [40] were used to compare ensemble averages between Para and non-disabled athlete groups. Impairment assessment measurements and performance measurements were analyzed separately using GLM. Partial eta squared (Pη2) values were calculated to estimate effect sizes and classified as small (0.01–0.059), moderate (0.06–0.137), or large (>0.138) [41].

Prior to statistical analysis to fulfil objective (3), data reduction was necessary. Firstly, Pearson correlations were performed to explore the relationship between all variables within the dataset for Para and non-disabled athlete groups separately. Where significant (*p* < 0.05) relationships (r > 0.7) between impairment measurements or between performance measurements were found, data reduction was performed to reduce autocorrelation between predictor variables. For the Para athlete group only, this was followed by Principal Components Analysis (PCA) with varimax orthogonal rotation (Kaiser normalization) to cluster the impairment assessment measurements into smaller groups of physical impairment types. Factor scores were calculated based on the Anderson–Rubin method. Kaiser–Meyer–Olkin and Bartlett’s test for sphericity were used to assess the sample size [40].

Stepwise linear regression was then used to identify impairment assessment measurements from data reduction that were predictors of performance measures. This was performed on both datasets (Para and non-disabled athletes) separately to assess validity. The ability of the regression model to be generalized to the eligible Para and non-disabled dressage populations was investigated using the Durbin–Watson statistic and Variance Inflation Factor (VIF) [40] to assess the assumption of independent errors and collinearity, respectively.

Significance was set to *p* < 0.05. All statistical analyses were performed in SPSS (version 28.0.1.1. (15), IBM Corp., Armonk, NY, USA).

## 3. Results

Descriptive statistics for impairment assessment measures for all athletes are presented in Table 2 and Table 3, and for performance measures in Table 4. Descriptive statistics for Para athletes separated by grade and non-disabled athletes separated by level are included in Appendix A. Significant differences (*p* < 0.05) between Para and non-disabled athletes for each measure are illustrated below their respective ensemble averages. Results of the GLMs found a significant main effect between Para and non-disabled athletes for impairment measures: F_(9)_ = 3.181, *p* = 0.038, and Pη^2^ = 0.998. No significant main effect in performance measures between Para and non-disabled athletes (F_(16)_ = 2.051, *p* = 0.093, and Pη^2^ = 0.585) was found. For performance measures, two grade I athletes did not ride the trot section of the test, and data are missing for head stability measures from two athletes (one Para grade I athlete and one non-disabled PSG athlete) due to sensor malfunction.

Pearson correlations performed on both datasets to reduce autocorrelation are presented in Appendix A. For both Para and non-disabled datasets, highly significant correlations were found for all peak and average HHD values (all above r = 0.95, *p* < 0.0001), and many of the strength measures were significantly correlated (r > 0.7, *p* < 0.01), except for hip external rotation. HHD measures that were chosen for further analysis were based on having fewer significant correlations to other HHD measures and included measures of trunk, left and right, upper, and lower limb strengths.

PCA of Para athlete impairment assessment measurements that were selected following data reduction (11 measurements—Table 5) resulted in a reduction in dimensions to three components, which explained 82.6% of the variance. Sampling adequacy was verified (Kaiser–Meyer–Olkin individual measurement values were all >0.69, and Bartlett’s test for sphericity χ^2^
_(55)_ = 210.750, *p* < 0.001 indicated that correlations were sufficiently large). The rotated component matrix is included (Table 5). Principal components were named based on the results of the analysis as PC1 = Strength, PC2 = Sitting Function (including coordination during sitting), and PC3 = Tone. The results of the analysis are illustrated in Figure 4.

Stepwise linear regression identified six impairment measures that could significantly (*p* < 0.05) predict performance in the Para athlete group (Table 6). These impairment measures could predict between 19% and 35% of the impact of impairment on performance, and only one predictor was found for each model. FIST and SARA could not be included in the regression analysis for the non-disabled group due to a ceiling effect. Two of the performance measures (TDS Trunk and TDS Pelvis) were predicted by an alternative impairment measure (TIS) in the non-disabled group. The Durbin–Watson statistic was between 0.999 and 2.252 for the Para athlete analysis, indicating that independent errors were not evident in the analysis. VIF values for each model and excluded variables were 1 to 1.666 in the Para group and 1 to 6.855 in the non-disabled groups, indicating that collinearity was not evident in the analysis. The results of the regression analysis are provided in Table 6 and illustrated in Figure 4. Trunk and pelvis dynamic symmetry patterns are shown in Figure 5.

## 4. Discussion

This study forms the final work package of the overarching research project commissioned by the FEI to investigate the impact of impairment on performance in Para dressage. These data provide original values for Para and non-disabled athletes from which other athletes can be compared. The dynamic symmetry of the trunk during simulated walking was significantly different between Para and non-disabled athlete groups, which partly confirmed the first hypothesis. Although average coordination variability in the Para athlete group was greater than that in the non-disabled group, the difference in coordination variability between Para athletes was large, so this was not distinctly measurable between groups. Coordination variability during simulated riding was independent of muscle strength in the Para athlete group, but in the non-disabled group, coordination variability was linked to trunk rotation and hip external rotation strength. As such, hypothesis two could not be accepted. For hypothesis three, performance was impacted more in athletes with activity limitations that affect the trunk and pelvis, which was evident in performance measurement relationships with FIST, SARA, and TIS that were grouped into the principal component related to sitting function. Hypothesis three was therefore accepted.

### 4.1. Impairment Measures

The validated clinical impairment assessment tools used in this study are currently used or have been investigated for use in other Para sports. For example, the Ashworth or Modified Ashworth Scale is commonly used in Para classification, and MAS was used to assess the effect of kinesio tape on stretch reflex in Para swimmers [42]. In another study, HHD was used to assess trunk strength in Para swimmers [18], but it was concluded that HHD consistently underestimated trunk strength in this population. SARA is used in Para archery and Para cycling classifications, and similar items to SARA are used in Para volley. SARA is a useful measure of impaired coordination in people with ataxia [12], and items from SARA are considered an appropriate measure of balance for the assessment of people with multiple sclerosis and cerebellar ataxia [43]. To the authors’ knowledge, the FIST has not been used as an impairment assessment tool in Para sport previously, but this tool also has sport specificity for Para dressage and was selected as a potential tool for identifying the impact of impairment on performance [12]. The FIST measures static, proactive, and reactive balance as well as sensory integration, and it has been used in populations with balance dysfunction, neurologically impaired patients, and individuals with multiple sclerosis [44]. A ‘ceiling effect’, which is a term used to indicate when a high proportion of participants achieve a maximum score, was evident in both the SARA and FIST assessment tools for the non-disabled group. Some grade 4 and 5 athletes also achieved a maximum score. Maximum scores have been reported previously for FIST in adults with sitting balance dysfunction [45] and for SARA in a large group of spinocerebellar ataxia patients covering the entire range of disease severity [46].

It is unsurprising that R-MAS identified impairments in the Para athletes, as R-MAS and MAS are widely recognized as reliable tools for quantifying muscle tone [47,48]. In this study, R-MAS was chosen over MAS due to reported improvements in reliability, such as for the measurement of upper limb muscle spasticity in patients with hemiparesis [49]. For both R-MAS and HHD, the joints chosen for assessment were based on their applicability to dressage riding performance [7,8,12]. For R-MAS, scores ranged from minimum to maximum, with eleven Para athletes across a range of functional profiles obtaining positive scores. Thus, findings suggest that HHD and R-MAS could be useful impairment assessments for Para dressage athletes, particularly when they are applied in a way that reflects rider position. However, non-eligible impairments can also be identified using these tools, as illustrated by one non-disabled athlete who had an increase in resistance at the end of ROM in one hip, which was attributed to a previous injury and weak hip stabilizers.

TIS is also a validated tool to assess sitting function, and indeed, in the Para athlete group, TIS was included in the sitting function principal component. Several of the non-disabled participants did not reach a perfect score of 23 for the TIS. This was principally due to the asymmetrical rotation of the upper and lower trunks when executing the dynamic coordination test items [14]. Similar findings have been reported [50] when comparing healthy participants to stroke patients. Healthy participants obtained maximum scores for static balance and achieved more than two-thirds of the available scores for dynamic balance and coordination items [50]. The lower scores among non-disabled athletes may be indicative of a sport-specific deficit in the dressage population.

For the HHD measures, the difference in strength between Para and non-disabled athletes was partly due to neuromuscular deficits in Para athletes but also because some did not have use of their lower limbs, resulting in a score of zero for some measurements. Strong correlations were found between shoulder and elbow measures in the Para athlete group, indicating comparable upper limb strength within most athletes. Hip HHD relationships were somewhat different, but this was in part due to the zero scores. In non-disabled athletes, trunk strength and hip strength measures were more strongly correlated. Interestingly, both groups of dressage athletes produced lower scores than previously reported normative data in non-disabled and patient populations [51]. The difference in gender could explain some of the discrepancies between studies, as [51] were mainly male whereas this study included mainly female athletes, but several other factors have been identified that limit the comparison of HHD measurements between studies. These include, but are not limited to, HHD placement, differences between devices, test position, strength of the tester, age range of participants, and the physical characteristics of the participant [52].

### 4.2. Performance Measures

Identifying suitable, quantifiable measures of Para dressage performance is challenging, given the multifactorial nature of this sport. Previous studies have investigated attributes of rider skill in non-disabled dressage athletes. However, findings from these studies are often confounded by small sample sizes, differing methodologies, and samples of athletes with a range of skill levels, which limits the ability to perform a meta-analysis [7]. In some studies, clear differences have been identified between experienced and novice athletes; in others, the strategies used by the athlete to influence the horse are reported to be highly individual [53] and can vary within an individual across different gaits and speeds [25]. The performance measurements used in this study were developed from previous studies that found differences between rider level and/or judged dressage scores [24,36,54,55,56,57,58,59]. Results indicated that, overall, the Para athletes’ performance was not significantly different from the non-disabled athletes’ performance when all performance measurements were included in the model. Similarities in ensemble means for some performance measures indicated that elite Para athletes can develop comparable skills to non-disabled athletes despite their impairments. As the specificity of a task influences trunk control [60], the predictability of the simulator reduced the demands of the postural control task. Although this may encourage the use of horses to conduct performance analysis in Para dressage athletes, motion variability between horses [61] limits the potential to compare between athletes.

Two notable measures reached statistical significance. The first was simulator rotation signal power, and the second was trunk dynamic symmetry. Frequency domain analysis of horse trunk acceleration has been reported previously [31], but the diversity of acceleration patterns produced by different horses limited the ability to detect differences between skill levels of the athletes. The advantage of using the simulator was the predictability of trunk motion if unperturbed. That said, the variability between the non-disabled athlete group skewed the data, as three of the athletes in this group notably restricted simulator trunk rotation compared to most of the group. As no correlations were identified between trunk power and other variables in the non-disabled athlete outliers, absorption of this energy was expected to occur either through undetected isometric muscle activity or due to lower limb function that was not quantified.

When riding dressage, the trunk of the athlete should remain closely aligned with the vertical, which requires the ability to resist accelerations and decelerations of the horses’ COM [62]. The walk is less challenging in this regard due to the relatively small changes in horse COM motion as a consequence of the slow speed and lack of suspension phases [63]. Previous studies have found no differences in trunk inclination between athlete skill levels at walk [30], although between athlete variation in saddle force profiles is evident [64] and has been linked to pelvis kinematics [65]. A walk on the simulator produced lateral motion similar to an overground walk, which is also described as more predictable [59]. Consequently, one might expect COM patterns of the pelvis and trunk in the horizontal plane to be equally predictable in elite dressage athletes unless they were impaired. Trunk control is commonly evaluated in sports and clinical studies [60,66], and unstable sitting tasks are used in classification systems in other Para sports [67,68,69,70]. In sitting balance tasks, the ability for the trunk to remain stable can be facilitated by reducing the degrees of freedom of the unstable surfaces, which has been demonstrated in people with cerebral palsy [71]. In the current study, the simulator hydraulic actuators translate in three dimensions at different frequencies for each gait [25], which produces predictable 3D rotational motion of the simulator trunk. The variability in trunk motion in the Para athletes in this study may be illustrative of individual differences in trunk control strategies for remaining stable during the translations and rotations of the simulator trunk. That said, the average trunk motion patterns were more asymmetric in Para athletes compared with non-disabled athletes, and since the Para athletes were known to have a high skill level, trunk asymmetry was attributed to limitations in functional capacity.

Variability in trunk position increases the demand on the upper arms to moderate the distance between the wrists and the bit [72], posing an added level of coordination complexity. As trunk dynamic symmetry is a predictor of the collective score for posture in non-disabled dressage [23] and consistency and lightness in rein tension are associated with behavioral and rideability judged traits [72,73,74], trunk dynamic symmetry was an important measure of performance limitation in Para dressage.

### 4.3. The Relationship between Impairment and Performance

In other Para-sports, studies have demonstrated the extent to which physical impairments may affect performance and used that information to support the inclusion of sport-specific impairment assessment measures for classification [75]. Para dressage is compounded by the range of eligible impairments and by the effect of the horse both individually and as a component of the horse-athlete dyad [7]. Many non-disabled dressage athletes have some limitations in functional capacity attributed, for example, to low back pain [76,77], inherent or acquired asymmetry [78], or other clinical conditions that are not eligible for Para status. This compounds the difficulty of identifying moderate relationships between impairment and performance measures. Even so, relationships between impairment measures and performance measures were found in the Para athlete group that were not evident in the non-disabled group. These involved three of the measures in the principal component of sitting function (including coordination): FIST, SARA, and TIS, and the one muscle tone measure, R-MAS.

The strength of the association between sitting function measures and performance measures was more evident in athletes with greater severity of impairments. For wheelchair users, movement restrictions from the chair may reduce strength in the trunk lateral flexors [71], limiting their ability to perform the lateral bending and reaching tasks in FIST and TIS. Lateral reaching requires optimal movement control of the trunk lateral flexors to keep the pelvis stable while shifting the COM close to the limits of stability [79]. Items in both tests require lateral pelvic lifting, which involves activation and control of trunk, pelvic, and abdominal muscles that contribute to core stability [80]. Although most studies to date have measured flexor and extensor muscle activity [81,82,83,84,85], one study reported high magnitudes of muscle activity from the internal oblique abdominus in older non-riders [82] and another between novice and advanced jockeys [86], highlighting the involvement of other core stabilizers. Greater mediolateral amplitude and velocity of the center of pressure have been measured during riding in a group of cerebral palsy participants with five years of therapeutic riding experience compared to controls, also illustrating a deficit in mediolateral trunk control due to impairment [87].

The FIST also includes forward-reaching tasks and picking up an object from the floor that involve activation and control of the erector spinae and rectus abdominis. The authors of [71] suggested that people with cerebral palsy who have better trunk control in dynamic conditions have more skills and proficiency in performing trunk flexion-extension tasks. In addition, a lack of movement control in the trunk lateral flexors is reported to result in excessive flexion of the upper torso against a posteriorly tilted pelvis [79]. Coordinated activation of the trunk flexors and extensors is essential for damping and postural control of the trunk during riding [81,82]. To compete in dressage, athletes must learn to stabilize their pelvis and trunk laterally, as well as anterior-posteriorly, while allowing their pelvis to move with the horses’ motion. For elite Para athletes in this study, deficits in picking up an object from the floor impacted their ability to position their trunk over their pelvis when riding the simulator. Consequently, FIST is likely to be useful as a functional assessment of the performance demands expected of Para dressage riders.

More experienced athletes have better control of their head movements [56,57], which is an advantage for performing gaits and movements precisely within the confines of a dressage arena. To test head stability, the study protocol included four downward transitions from walk to halt that were controlled by the simulator operator. The athletes were aware of this prior to performing the test; however, cues were not provided during the test; hence, the halts induced an anterior perturbation that the athletes had to control. When perturbed in this way, the trunk and upper body move as an inverted pendulum, and lower back motor control is involved in head-in-space stabilization [88]. Feedback from visual, vestibular, proprioceptive, and tactile afferents is used to coordinate hypaxial and epaxial muscle activation to stabilize the body [89]. That said, visual processing time results in a latency time delay, so visual feedback only contributes at low frequencies [88], and more skilled athletes are less dependent on visual input [57]. An association was found between head stability and trunk symmetry and the overall SARA score for Para athletes, which is not surprising because SARA represents elements of a standard neurological examination [46].

Although pelvic symmetry was not different between groups, R-MAS could predict 30% of the variation in Para athletes. All except one of the athletes with positive scores had increased tone in one or both hips. In both groups of athletes, other physical impairments, functional deficits, or asymmetries influenced dynamic pelvic symmetry. For example, dominant lateral flexion and rotational asymmetry in both the trunk and pelvis were reported by [90] in a group of athletes with proficient skill in riding sitting trot. Lateral pelvic tilt during standing is frequently reported [78,81,86]. Thus, the confounding effects of non-eligible impairments add to the complexity of the analysis. However, pelvic motion on the simulator should follow a predictable symmetric pattern, and variations in pelvic motion might assist in identifying specific activity characteristics that are not necessarily connected to eligible impairments.

### 4.4. Limitations

Limitations of the current study include the relatively small sample size, which was influenced by difficulties in recruiting classified elite Para dressage athletes from outside of the UK due to travel restrictions and rising costs caused by the COVID-19 outbreak during the testing period. Further, the specific level of rider determined by the inclusion/exclusion criteria required for the testing reduced the pool of eligible athletes. While the impairment assessment tools demonstrate useful application in the investigation of the relationship between impairment and performance of Para dressage and non-disabled riders, further assessment of the usability of these tools or elements of tools in the field is required.

The Para athletes who volunteered to participate in this study were from an elite population which reduced the confounding effect of skill level. Athletes at the highest level focus on improving and maintaining their physical capacity to compete successfully in the sport. This was recognized early in the development of the sport by [91], who concluded that success was influenced by impairment and the amount of training, preparation, and rider talent. Horse riding itself induces various physiological changes that are beneficial [92] and is reported to improve muscle strength in the elderly [93]. As such, the confounding effects of training may have impacted the results.

The use of the simulator to provide a standard method of assessing an athlete was, on balance, considered the preferred method over the use of live horses for reasons that have already been discussed. However, it also has inherent limitations. During pilot testing, lower-level Para and non-disabled athletes assisted in the development of the protocol. From this work, it was evident that the sensors on the simulator could not be used to engage the simulator in forward motion by some athletes (due to their position), and some athletes failed to keep the simulator in the virtual arena, which could stop the program from running. The dressage test and method used to control speed were designed with this in mind to ensure that, as a minimum, all walk elements could be performed by all athletes. For some athletes, the lack of acceleration and deceleration that they would normally experience on a horse overground made the task less challenging. In addition, the regularity of the simulator motion may have reduced the difficulty in maintaining stability compared to the athletes’ horse, impacting trunk and pelvis symmetry and coordination variability measurements. That said, differences in performance between Para and non-disabled athletes were identified because the method was standardized.

Prior to conducting the principal component analysis and stepwise linear regression, the number of HHD measures was reduced from thirty-six to seven due to the strong and very strong correlations influencing the outcome. Our choice of measures was based on ensuring that measures from the trunk, upper, and lower limbs were included in the analysis. Although we provided a justification, we acknowledge that the HHD measures included in the analysis may not have captured all activity limitations related to eligible impairments in the Para athlete group.

## 5. Conclusions

This study focused on investigating the impact of impairment on performance in elite Para dressage athletes. Impairment assessment measurements of strength, sitting function, and tone detected deficits in the Para athlete group, which impacted dynamic trunk symmetry during a simulated dressage performance. Three impairment assessments were identified that could predict between 19% and 35% of the impact of impairment on performance in Para dressage athletes but not in non-disabled athletes. Of these, FIST and SARA impairment assessments most strongly identified impairments related to sitting function, and R-MAS uniquely identified impairments related to muscle tone. These original findings provide the first sport-specific scientific evidence that can be used to assist the FEI in the review and refinement of the current classification system.

## Figures and Tables

**Figure 1 animals-13-02785-f001:**
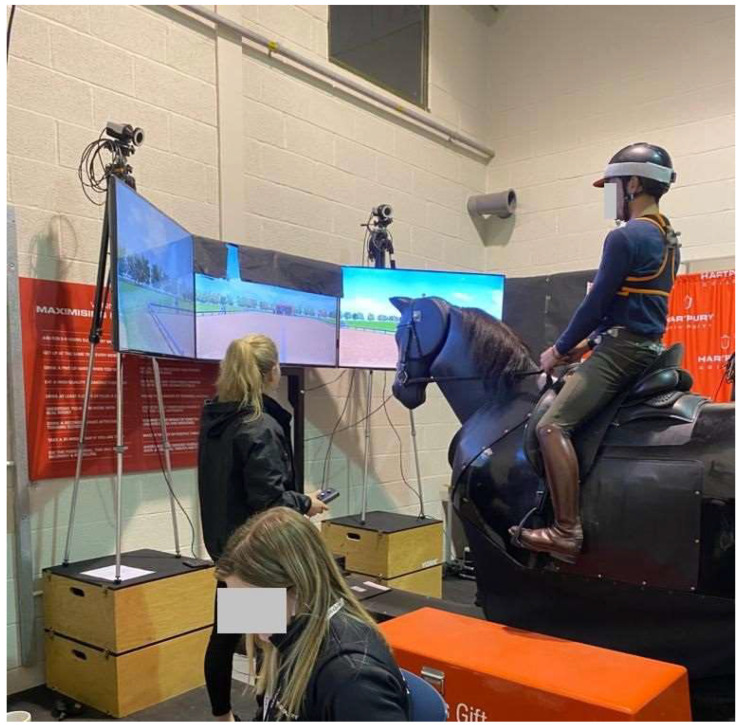
Simulator set-up.

**Figure 2 animals-13-02785-f002:**
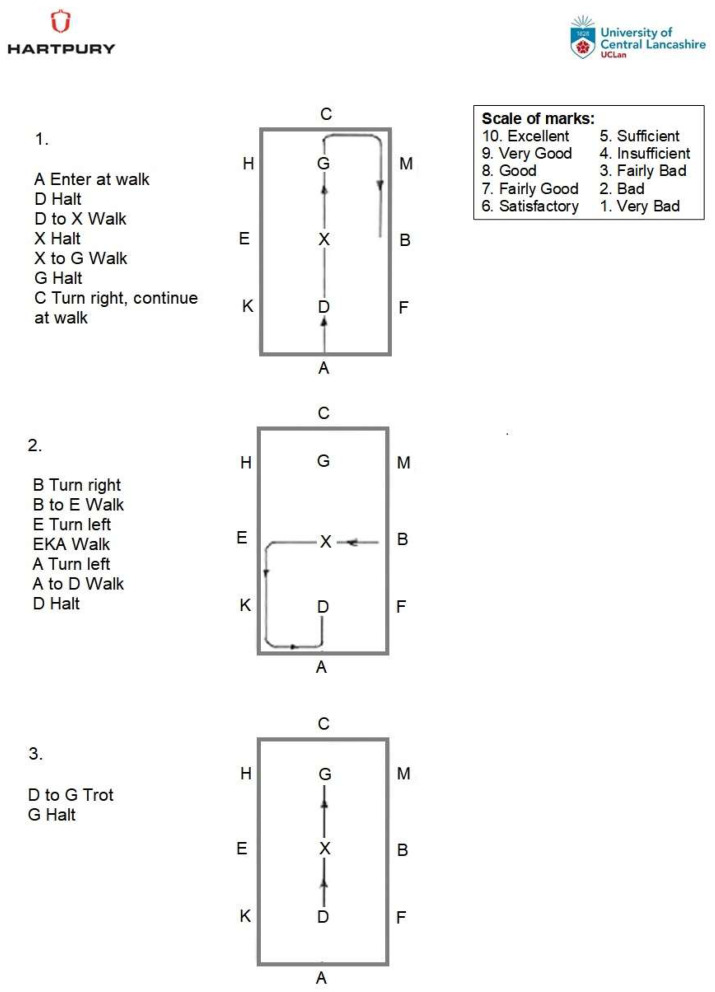
Custom dressage test.

**Figure 3 animals-13-02785-f003:**
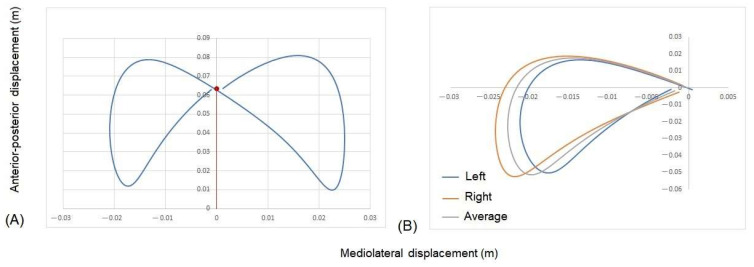
An example of the method used to develop dynamic symmetry measurements for a non-disabled athlete. (**A**) Dynamic pattern of motion of the pelvis center of mass (COM) in a horizontal plane. The red dot is the midpoint of the pattern. The symmetry vector is represented by the red line from location 0.0. (**B**) Dynamic symmetry from the midpoint was compared by folding the right side over the left and determining the difference in distance between the left and right sides at each normalized time point.

**Figure 4 animals-13-02785-f004:**
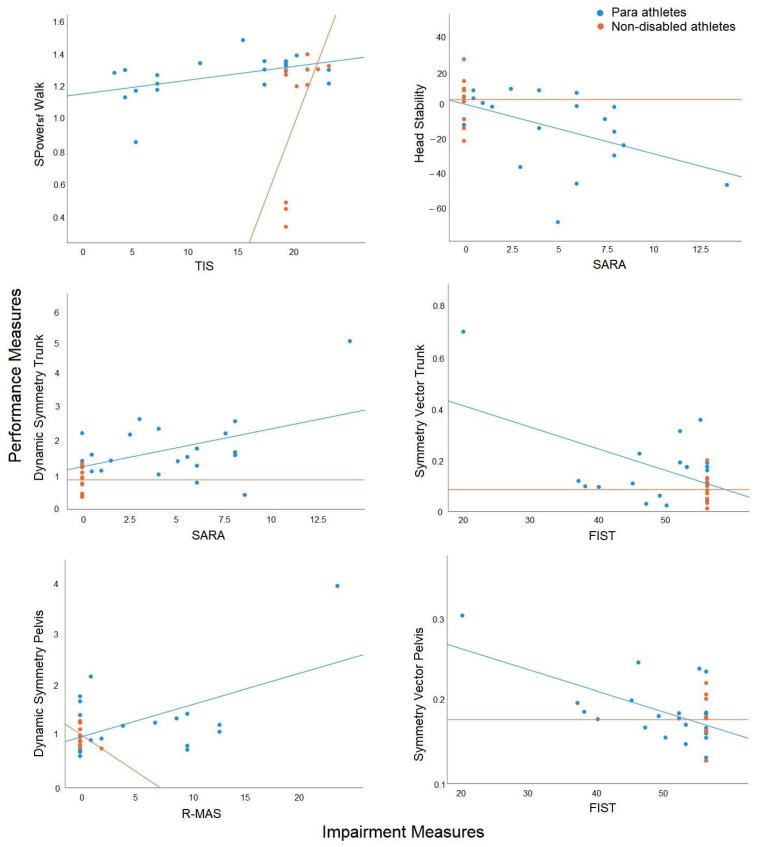
Results of regression analysis showing the significant (*p* < 0.05) relationships between impairment and performance measures for the Para athlete group (blue), contrasted by the non-significant (*p* > 0.05) relationships in the non-disabled group (orange).

**Figure 5 animals-13-02785-f005:**
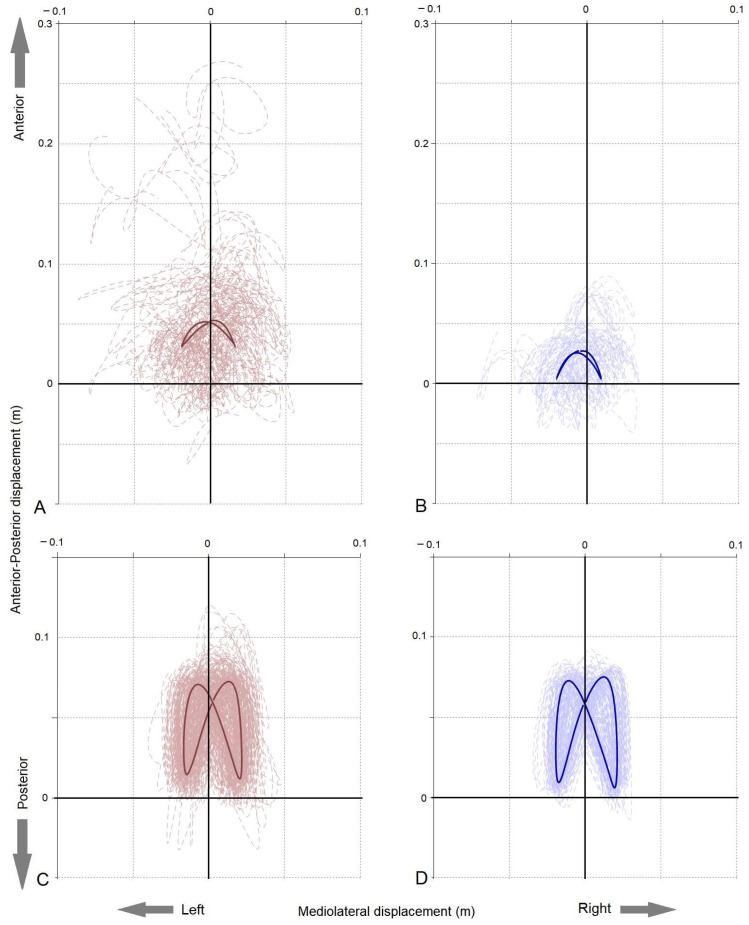
Average (solid-bold) and individual stride (dashed) centre of mass (COM) paths in the horizontal plane (m) of the trunk and pelvis that were used to develop dynamic symmetry measures for Para and non-disabled athlete groups. (**A**) Para athlete group trunk, (**B**) non-disabled athlete group trunk, (**C**) Para athlete group pelvis, and (**D**) non-disabled athlete group pelvis. The crosshairs are located at 0.0, which is the COM position of the pelvis during the static trial.

**Table 1 animals-13-02785-t001:** Demographic data for participants, grouped by Para and non-disabled dressage athletes. The number of athletes (n) is shown for each demographic category.

			**n**
Para athletes (n = 21)	Gender	Male	1
		Female	20
	Age: Mean (SD) years	41.1 (13.2)	
	Impairment	Impaired muscle power	8
		Ataxia	5
		Athetosis	2
		Hypertonia	1
		Dystonia	1
		Impaired Range of motion (ROM)	2
		Visual Impairment	2
	Current Grade (Sports Class)	1	6
		2	3
		3	4
		4	6
		5	2
	Classification Status	Confirmed (C)	16
		Review (R)	5
	Competition Information	Number of International Competitions (1 January 2018 to 31 December 2022)	177
		Placings at Major Competitions	81
		Number of Medals	59
Non-disabled athletes (n = 11)	Gender	Male	2
		Female	9
	Age: Mean (SD) years	35.8 (13.7)	
	Competition Level	Prix St. Georges	6
		Grand Prix	5

**Table 2 animals-13-02785-t002:** Average scores (mean (standard deviation)) grouped for Para athletes and non-disabled athletes with corresponding *p*-values between the two groups below for impairment assessment measures. Function In Sitting Test (FIST), Scale and Assessment of Rating for Ataxia (SARA), Trunk Impairment Scale (TIS), and Re-Modified Ashworth Scale (R-MAS). Peak handheld dynamometry measures (N) for the trunk are presented. Note: %COV for SARA and R-MAS was calculated by dividing the SD by the difference between the mean score and the total available score. Number of athletes (n).

						Trunk Handheld Dynamometry (Peak Values)					
Group	n	FIST	SARA	TIS	R-MAS	Flexion	Extension	Rotation to Left	Rotation to Right	Left Lateral Flexion	Right Lateral Flexion
Para	21	49.00 (9.07)	4.74 (3.60)	13.43 (7.10)	4.95 (6.52)	60.8 (31.6)	75.4 (42.0)	54.1 (26.0)	51.9 (26.8)	66.3 (31.1)	61.8 (27.4)
%COV		19	24	53	34	52	56	48	52	47	44
Non-disabled	11	56.0 (0.0)	0.0 (0.0)	20.27 (1.42)	0.18 (0.60)	105.6 (21.5)	141.8 (42.6)	90.8 (20.0)	87.1 (20.1)	121.2 (30.1)	118.6 (23.7)
%COV		0	0	7	3	20	30	22	23	25	20
*p*-value		0.017	<0.001	0.004	0.023	<0.001	<0.001	<0.001	<0.001	<0.001	<0.001

**Table 3 animals-13-02785-t003:** Average scores (mean (standard deviation)) grouped for Para athletes and non-disabled athletes with corresponding *p*-values between the two groups below for peak handheld dynamometry (HHD) measures (N) for the limbs. Adduction (Add), External Rotation (Ex Rot), and Extension (Extn). Number of athletes (n).

		Hip (Peak Values)				Shoulder (Peak Values)				Elbow (Peak Values)			
Group	n	Add Right	Add Left	Ex Rot Right	Ex Rot Left	Extn Right	Extn Left	Ex Rot Right	Ex Rot Left	Flexion Right	Flexion Left	Extn Right	Extn Left
Para	21	54.4 (53.8)	48.7 (48.7)	31.7 (29.0)	32.3 (28.1)	59.9 (38.1)	61.4 (34.4)	43.5 (20.4)	49.6 (24.7)	52.7 (29.7)	53.0 (34.0)	52.1 (33.3)	58.0 (36.8)
%COV		99	100	91	87	64	56	47	50	56	64	64	63
Non-disabled	11	119.5 (30.9)	117.5 (33.6)	76.8 (18.1)	79.1 (24.9)	113.3 (29.2)	110.1 (30.8)	85.3 (23.4)	88.7 (20.2)	94.7 (25.7)	101.2 (28.1)	90.7 (22.4)	101.6 (25.9)
%COV		26	29	24	31	26	28	27	23	27	28	25	25
*p*-value		<0.001	<0.001	<0.001	<0.001	<0.001	<0.001	<0.001	<0.001	<0.001	<0.001	0.002	0.002

**Table 4 animals-13-02785-t004:** Average scores (mean (standard deviation)) grouped for Para athletes and non-disabled athletes with corresponding *p*-values between the two groups below for performance measures. These are as follows: three-dimensional simulator trunk rotation signal power (SPower_sf_) (deg^2^.s) and harmonic ratio (SRatio_sf_) at the stride frequency during simulated walk and trot; head stability (%); dynamic symmetry and symmetry vector for the trunk (Sym Trunk, SVector Trunk) and pelvis (Sym Pelvis, SVector Pelvis) (normalised to trunk length and pelvis width, respectively) during walk; average absolute deviation of detrended coordination variability of 3D trunk to pelvis rotation (deg) during simulated walk (DVar walk) and trot (DVar Trot). Number of athletes (n). Note: for head stability, data from one Para athlete and one non-disabled athlete were missing.

Group	n	SPower_sf_ Walk	SRatio_sf_ Walk	SPower_sf_ Trot	SRatio_sf_ Trot	Head Stability	Sym Trunk	SVector Trunk	Sym Pelvis	SVector Pelvis	DVar Walk	DVar Trot
Para	21	1.27 (0.12)	1.38 (0.09)	0.22 (0.18)	3.93 (4.34)	−13.08 (22.23)	1.88 (0.98)	0.17 (0.15)	1.29 (0.73)	0.19 (0.04)	13.18 (8.23)	7.31 (9.64)
%COV		9	6	82	110	170	52	88	57	21	62	132
Non-disabled	11	1.06 (0.41)	1.36 (0.05)	0.20 (0.05)	6.42 (11.61)	3.14 (14.10)	0.91 (0.37)	0.09 (0.05)	1.02 (0.20)	0.18 (0.03)	8.68 (2.86)	5.76 (2.01)
%COV		39	4	25	181	449	41	56	20	17	33	35
*p*-value		0.027	0.401	0.529	0.920	0.084	<0.001	0.071	0.259	0.636	0.116	0.577

**Table 5 animals-13-02785-t005:** Rotated Component Matrix from Principal Component Analysis of impairment assessment measurements. The rotation converged after seven iterations. Shading is used to highlight the impairment assessments that cluster on the same principal component based on their factor loadings.

	Component		
	PC1 Strength	PC2 Sitting Function	PC3 Tone
FIST	0.221	0.860	0.089
SARA	0.404	−0.799	0.247
TIS	0.423	0.720	0.007
R-MAS	−0.194	0.024	0.852
HHD Trunk Flexion	0.890	0.060	0.181
HHD Trunk Rotation toward left	0.799	0.468	−0.081
HHD Trunk Rotation toward right	0.800	0.494	0.111
HHD Right Hip External Rotation (seated)	0.417	0.639	0.456
HHD Left Hip External Rotation (seated)	0.392	0.660	0.412
HHD Right Shoulder Extension	0.943	0.019	−0.163
HHD Left Shoulder Extension	0.916	0.216	−0.153

**Table 6 animals-13-02785-t006:** Results of regression analysis for Para and non-disabled (ND) athletes. Performance measures are as follows: three-dimensional simulator trunk rotation signal power (SPower_sf_) (deg^2^.s) and harmonic ratio (SRatio_sf_) at the stride frequency during simulated walk; head stability (%); dynamic symmetry and symmetry vector for the trunk (Sym Trunk, SVector Trunk) and pelvis (Sym Pelvis, SVector Pelvis) (normalised to trunk length and pelvis width, respectively) during walk; and average absolute deviation of detrended coordination variability of 3D trunk to pelvis rotation (deg) during simulated walk (DVar walk). Impairment measures are peak trunk rotation to the left (TrRot L) and peak right hip external rotation (HipExRot R) (N). Shading is used where no predictors (NP) were found.

Performance Measure	Group	Regression Model and Predictor(s)	Standardized Beta Coefficients	R^2^	Significance	Durbin–Watson Statistic
SPower_sf_ Walk	Para	TIS	0.489	0.239	0.025	0.999
	ND	NP				
SRatio_sf_ Walk	Para	NP				
	ND	NP				
SPower_sf_ Trot	Para	NP				
	ND	NP				
SRatio_sf_ Trot	Para	NP				
	ND	NP				
Head Stability	Para	SARA	−0.481	0.232	0.032	1.841
	ND	NP				
Sym Trunk	Para	SARA	0.436	0.190	0.048	1.875
	ND	TIS	−0.636	0.404	0.035	1.568
SVector Trunk	Para	FIST	−0.511	0.261	0.018	2.064
	ND	NP				
Sym Pelvis	Para	R-MAS	0.545	0.297	0.011	2.252
	ND	TIS	−0.630	0.397	0.038	1.775
SVector Pelvis	Para	FIST	−0.590	0.348	0.005	1.272
	ND	NP				
DVar Walk	Para	NP				
	ND	TrRot L	−0.671	0.450	0.024	2.317
DVar Trot	Para	NP				
	ND	HipExRot R	−0.684	0.468	0.020	2.151

## Data Availability

An anonymized datasheet of the clinical and performance measures from this study is available on request at: https://uclandata.uclan.ac.uk/id/eprint/396.

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
