# Peer review of "Towards an Evidence-Based Classification System for Para Dressage: Associations between Impairment and Performance Measures"

_animals, 2023, doi:10.3390/ani13172785_

Round 1

Reviewer 1 Report

The growth of adaptive equine interactive sports over the past few years warrants the growth of research within this field, and thus, this manuscript is timely, if not well overdue, and of interest to readers. As such, it seems that the authors are trying to compensate for this lack of published work within this area, and thus, the introduction and discussion sections of the current manuscript well exceed the length of what is expected within one manuscript. It may be advised this work needs to be divided within a literature review and a research article as it seems much background information is needed in laying down the foundation for the findings within this study. Authors refer to a previously published scoping review focused within this topic area, however, scoping reviews can be limiting for potential in-depth discussion of a topic area depending on the questions explored within the scoping review and the number of associated literature found for these questions investigated within the review. As such, as the authors try to streamline the introduction and discussion sections within the current manuscript, they may look to moving some of this information to a literature review. For example, easily sections 4.4, 4.6, and 4.7 can be removed from the discussion section of the current manuscript, but content within these sections could be of value within a literature review. In any case, both the introduction and discussion sections within the current manuscript get off-course at times from the main objectives of this study and can seem repetitive making both sections substantially too long for the content presented within the results so that significant reductions are warranted. Nevertheless, while the introduction is too lengthy, a hypothesis statement needs to be added after the objectives statement at the end of the introduction.

Along with the introduction and discussion sections being well too long for the current manuscript, another limitation is the sample size in which the authors are commended for acknowledging this limit within the discussion section. Nevertheless, to fully understand the appropriateness of the sample size utilized, authors should indicate within the manuscript what are the current numbers of Para dressage athletes, particularly within each grade. See comments below concerning an Appendix table. Further, authors should keep in mind this study also collected data on Prix St George and Grand Prix dressage riders, and thus, what are the current numbers within this population as that should also be reported to reflect again appropriateness of sample size? A power analysis should be done to reflect whether sample size is appropriate for both groups, the Para athlete and the non-disabled athlete. Further, are numbers appropriate for each grade investigated or would this more represent a pilot study? The power analysis should assist in determining classification of the current study. In any case, current numbers of these groups should be given within the methods to reflect appropriate sampling. In reviewing sample size, it’s important to note that authors mention within line 864 that due to COVID-19 that sampling outside of the U.K. was limited. As such, were all of the participants, both Para athletes and non-disabled athletes, from the U.K.? If so, then, authors may be advised to adjust title and methods so that the sample populations for both groups were only from the U.K. Restricting to one geographical location may help to strengthen the power analysis, however, it may limit application in some ways to just those riders from that area. Therefore, adjust accordingly. If participants are from multiple locations, add in which countries that were represented by the participants either within Table 2 or elsewhere. See comments below.

Since the numbers are so low for participants, more specifics are needed concerning participants. While Table 1 is useful in exploring some specifics related to participants, another Table is needed within the Appendix that breaks down the information given in Table 1 for each partcipant. Specifically, per participant the following information should be given: age, gender, country of origin, grade if Para athlete or the level of rider if non-disabled rider, number of competitions/years of competition, and impairment if Para athlete. Since the authors repeatedly discuss within the discussion section how certain disabilities impact performance (i.e. lines 608, 610, 572, and 761) as it relates to assessments like TIS, FIST, and SARA, the specific diagnosed disability by the participant’s physician needs to be included for each participant within this Appendix table. Further, statistical analysis should explore whether the same is true within this study as to the impact of the disability to make sure that this can be ruled out for the current data. While the grading system does group together a variety of disabilities within each classification, previous research as given by the authors does indicate that specific disabilities can influence the variables measured, and as such, should be explored to ensure that is not a possibility. Also, within either the methods or within the Appendix, further details concerning the grading system is needed, potentially within a table format showing common disabilities/impairments that fall within each category and other details for understanding the differences between the grades. This added Appendix table may be an appropriate location to also include number of current participants within each grade as discussed in the previous paragraph of this review. Further, typically FEI has grade 1 divided into 1a and 1b, and as such, what was the breakdown within that classification for the sampled riders within grade 1? This should be addressed and determined if the division within grade 1 had any impact on results.

Within the methods section and even into the results, authors need to remember that there are two groups of participants, Para athletes and non-disabled athletes, and as such, specify specifically when saying “participants”, as to which ones are being referred to. For example, in line 297 authors mention “all participants”, but was that including the non-disabled participant? Most of the focus within the methods appears to be on the timing and protocol of assessments for the Para athletes, but that may be more because of the generalized use of “participants”. Again, authors need to be specific on which group is being referred to and further information concerning timing of assessments and procedures specific to non-disabled participants needs to be added or further clarified. If timing and procedures are exactly the same throughout for all aspects of the methods, then, reiterate that so that each step clearly indicates these similarities between the two groups. Further, while impairment assessments were done for the Para athlete, what medical history and clinical assessment was done to verify the lack of impairments for the non-disabled participants? How was a lack of disability confirmed? Even the authors mention within lines 734-737 that “many non-disabled athletes have some functional capacity” impacting performance, and thus, how was that ensured there wasn’t any type of unforeseen impairment that could impact data, especially with such low sample size? In addition, due to the sample size, addressing correlation strength as strong versus very strong mentioned within line 429 and within reporting results should be avoided. Strength of correlation analysis is often more subjective in nature, and instead, authors should just give the r value and let the readers interpret the strength of the values being presented.

For the results section, specifically Tables 2, 3, and 4, don’t include the combined Para athlete data, but instead, just keep the combined results of all Para athletes to the text and leave just the data presented per grade within these tables. Also, for the non-disabled participant data within these tables separate between Prix St George and Grand Prix riders, and then, when combining the two levels of dressage riders, present that within the text, not the table. Finally, within these tables, add either shading, border, or a row to divide between the Para athletes data for the five grades and non-disabled athletes for the two levels of dressage. These changes will assist in better visualizing the data presented within the tables. Further, Tables 5 and 6 need to indicate in either the titles or added notes what the shading represents. Figure 5 should be removed unless authors add in data for all participants instead of just examples. Again, similar to what was advised for the methods, authors need to indicate which group they are referring to within the results when they mention “participants” or “all participants”. Also, line 454, authors mention that head stability measures from two athletes was unavailable, but from what specific two groups was data missing from as with sample size so low for each grade a loss of data can have a substantial impact on results.

See comments and suggestions for authors.

Author Response

The growth of adaptive equine interactive sports over the past few years warrants the growth of research within this field, and thus, this manuscript is timely, if not well overdue, and of interest to readers. As such, it seems that the authors are trying to compensate for this lack of published work within this area, and thus, the introduction and discussion sections of the current manuscript well exceed the length of what is expected within one manuscript. It may be advised this work needs to be divided within a literature review and a research article as it seems much background information is needed in laying down the foundation for the findings within this study. Authors refer to a previously published scoping review focused within this topic area, however, scoping reviews can be limiting for potential in-depth discussion of a topic area depending on the questions explored within the scoping review and the number of associated literature found for these questions investigated within the review. As such, as the authors try to streamline the introduction and discussion sections within the current manuscript, they may look to moving some of this information to a literature review. For example, easily sections 4.4, 4.6, and 4.7 can be removed from the discussion section of the current manuscript, but content within these sections could be of value within a literature review. In any case, both the introduction and discussion sections within the current manuscript get off-course at times from the main objectives of this study and can seem repetitive making both sections substantially too long for the content presented within the results so that significant reductions are warranted.

Thank you for your comments. We have removed a large section from the introduction (originally lines 122 to 192), and sections 4.4, 4.6 and 4.7 as suggested. We will consider using this work as suggested to develop a literature review instead.

Nevertheless, while the introduction is too lengthy, a hypothesis statement needs to be added after the objectives statement at the end of the introduction.

We have added hypotheses to the end of the introduction, as requested. These are based on the text and references that were removed from the introduction.

Along with the introduction and discussion sections being well too long for the current manuscript, another limitation is the sample size in which the authors are commended for acknowledging this limit within the discussion section. Nevertheless, to fully understand the appropriateness of the sample size utilized, authors should indicate within the manuscript what are the current numbers of Para dressage athletes, particularly within each grade. See comments below concerning an Appendix table. Further, authors should keep in mind this study also collected data on Prix St George and Grand Prix dressage riders, and thus, what are the current numbers within this population as that should also be reported to reflect again appropriateness of sample size?

We acknowledged the limitations of the sample size for this study within the manuscript and adjusted the statistical analysis accordingly to compare between Para (n=21) and non-disabled (n=11) groups only. We have extracted information from the FEI classification database that provides athlete performance history to determine the number of athletes who were eligible for the Para athletes. From 100 registered athletes that represent the three countries who volunteered, 44 met the eligibility criteria, which was 48% of the population tested. The current worldwide online database has 559 athletes listed, of which 172 were eligible at the time that data was collected. Consequently, we tested just over 12% of the worldwide population of Para dressage athletes that were eligible. The non-disabled athletes were a control group and represent a small sample of the non-disabled dressage population. We have included additional information in the methods and limitations.

A power analysis should be done to reflect whether sample size is appropriate for both groups, the Para athlete and the non-disabled athlete. Further, are numbers appropriate for each grade investigated or would this more represent a pilot study? The power analysis should assist in determining classification of the current study. In any case, current numbers of these groups should be given within the methods to reflect appropriate sampling.

Thank you for your suggestion. We are sure you fully appreciate that relevant empirical data must be available in order to conduct a power analysis. We carefully considered this at the outset of the project, but empirical data is lacking that could be used to sensibly predict a sample size, as Para dressage athletes have never previously been tested in this way. Clinical tests are mainly reported in patient populations, which are not equivalent to elite Para athlete populations and very few studies have tested elite non-disabled dressage athletes. No studies were available that had investigated Para dressage athlete performance. For non-disabled athletes, in the performance scoping review we conducted the number of participants in studies ranged from 2 (1 Pro rider and 1 novice rider) to 25 (13 club riders and 12 Pro riders). Just to note, professional riders are not necessarily GP or Prix St Georges dressage athletes. Another study tested 7 elite dressage athletes. https://doi.org/10.3920/CEP150035. In a very recent study 19 high level (5 Prix St Georges and 14 GP athletes) were tested.

Several months before data collection began we undertook a pilot study with 7 lower level riders and one lower level Para athlete on the simulator. This pilot work, together with a day working with the UK classifiers was used (along with all previous studies) to develop the protocol for the main study. These data were not published but we have stated that this work was carried out in the introduction. As such, and due to our comments above, we do not consider this study to be a pilot study. In addition, we presented information from each grade as descriptives, but did not analyse it in this way. The analysis was only related to testing between Para and non-disabled athlete groups. We have considered your response here and feel that there is probably a need to split the tables and actually present the group data (i.e. Para athletes v non-disabled athletes) instead of the data by grade. We have included the data by grade and level as descriptive supplementary information.

In reviewing sample size, it’s important to note that authors mention within line 864 that due to COVID-19 that sampling outside of the U.K. was limited. As such, were all of the participants, both Para athletes and non-disabled athletes, from the U.K.? If so, then, authors may be advised to adjust title and methods so that the sample populations for both groups were only from the U.K. Restricting to one geographical location may help to strengthen the power analysis, however, it may limit application in some ways to just those riders from that area. Therefore, adjust accordingly. If participants are from multiple locations, add in which countries that were represented by the participants either within Table 2 or elsewhere. See comments below.

Please see above comments.

Since the numbers are so low for participants, more specifics are needed concerning participants. While Table 1 is useful in exploring some specifics related to participants, another Table is needed within the Appendix that breaks down the information given in Table 1 for each partcipant. Specifically, per participant the following information should be given: age, gender, country of origin, grade if Para athlete or the level of rider if non-disabled rider, number of competitions/years of competition, and impairment if Para athlete. Since the authors repeatedly discuss within the discussion section how certain disabilities impact performance (i.e. lines 608, 610, 572, and 761) as it relates to assessments like TIS, FIST, and SARA, the specific diagnosed disability by the participant’s physician needs to be included for each participant within this Appendix table.

Whilst we understand your reasons for requesting this information, we are unable to provide the detail you have requested because individual participants can be identified from such information. This would breach our ethical and GDPR regulations. The information included in Table 1 is what we are able to provide about the athletes. The reason for discussing specific disabilities within the discussion is that these disabilities are found within the Para dressage population, being eligible impairments. We have revisited the discussion and modified some of the text with your comments in mind.

Further, statistical analysis should explore whether the same is true within this study as to the impact of the disability to make sure that this can be ruled out for the current data. While the grading system does group together a variety of disabilities within each classification, previous research as given by the authors does indicate that specific disabilities can influence the variables measured, and as such, should be explored to ensure that is not a possibility.

We included information as part of the discussion related to specific disabilities/impairments as these are eligible impairments in Para dressage and we have few comparisons to draw upon outside of clinical literature that generally studies one or more than one clinical condition. This study was not designed to evaluate specific impairments, mainly because, as you have stated above, athletes are not grouped by impairment types in Para dressage.

Also, within either the methods or within the Appendix, further details concerning the grading system is needed, potentially within a table format showing common disabilities/impairments that fall within each category and other details for understanding the differences between the grades. This added Appendix table may be an appropriate location to also include number of current participants within each grade as discussed in the previous paragraph of this review. Further, typically FEI has grade 1 divided into 1a and 1b, and as such, what was the breakdown within that classification for the sampled riders within grade 1? This should be addressed and determined if the division within grade 1 had any impact on results.

The grading system for FEI classification is very detailed. The document describing eligible impairments, minimum eligibility criteria, the grading system and the classification process is available on the FEI webpages. References 4 and 6 contain all of the details related to classification, which is the most up to date official information. Previously grade 1 was broken down into grade 1a and 1b, but this was not the case at the time that we designed the study. Athletes are classified into 5 grades (see lines 48 to 60 of the introduction).

Within the methods section and even into the results, authors need to remember that there are two groups of participants, Para athletes and non-disabled athletes, and as such, specify specifically when saying “participants”, as to which ones are being referred to. For example, in line 297 authors mention “all participants”, but was that including the non-disabled participant? Most of the focus within the methods appears to be on the timing and protocol of assessments for the Para athletes, but that may be more because of the generalized use of “participants”. Again, authors need to be specific on which group is being referred to and further information concerning timing of assessments and procedures specific to non-disabled participants needs to be added or further clarified. If timing and procedures are exactly the same throughout for all aspects of the methods, then, reiterate that so that each step clearly indicates these similarities between the two groups.

Apologies, we have changed participants to Para and non-disabled athletes to ensure it is clear which group we are referring to through the methods.

Further, while impairment assessments were done for the Para athlete, what medical history and clinical assessment was done to verify the lack of impairments for the non-disabled participants? How was a lack of disability confirmed? Even the authors mention within lines 734-737 that “many non-disabled athletes have some functional capacity” impacting performance, and thus, how was that ensured there wasn’t any type of unforeseen impairment that could impact data, especially with such low sample size?

All participants completed a Par Q prior to commencing the study. For non-disabled athletes, medical history was discussed during their clinical assessment and any relevant medical history related to their clinical assessment recorded (which is included in the discussion where relevant as you have indicated). Sport-specific injuries are common in all sports, and dressage is no different in that context. In other studies, a high prevalence of back pain is reported in higher level dressage athletes due to the demands of riding. Confounding effects of non-eligible impairments are therefore potentially evident in Para and non-disabled dressage athletes in all studies. We have included additional information in the limitations related to this.

In addition, due to the sample size, addressing correlation strength as strong versus very strong mentioned within line 429 and within reporting results should be avoided. Strength of correlation analysis is often more subjective in nature, and instead, authors should just give the r value and let the readers interpret the strength of the values being presented.

We have removed the correlation strength from line 429 and other places and included the significance instead.

For the results section, specifically Tables 2, 3, and 4, don’t include the combined Para athlete data, but instead, just keep the combined results of all Para athletes to the text and leave just the data presented per grade within these tables. Also, for the non-disabled participant data within these tables separate between Prix St George and Grand Prix riders, and then, when combining the two levels of dressage riders, present that within the text, not the table.

Thank you for your suggestion. The problem with doing this is that the GLM statistics are between the two groups (Para athletes and non-disabled athletes). As such, we feel it is more important to present the group findings and the statistics between the two groups than the information by grade/ level. We have however included the data broken down by grade/level in an appendix.

Finally, within these tables, add either shading, border, or a row to divide between the Para athletes data for the five grades and non-disabled athletes for the two levels of dressage. These changes will assist in better visualizing the data presented within the tables. Further, Tables 5 and 6 need to indicate in either the titles or added notes what the shading represents.

Thank you for your suggestions. We have endeavored to improve the readability of these tables.

Figure 5 should be removed unless authors add in data for all participants instead of just examples.

We have developed a figure to include the mean and SD of all athletes in each group instead of only including examples.

Again, similar to what was advised for the methods, authors need to indicate which group they are referring to within the results when they mention “participants” or “all participants”.

Apologies, we have tried to ensure that it is always clear who we are referring to throughout the paper.

Also, line 454, authors mention that head stability measures from two athletes was unavailable, but from what specific two groups was data missing from as with sample size so low for each grade a loss of data can have a substantial impact on results.

We have included specific information about the loss of data here.

Reviewer 2 Report

animals-2522591

The paper based on the huge project is very interesting, detailed, and professional. Some points should be taken into consideration:

1. it is a very good report of the project, however, it seems too long for a single paper; the Authors should review if they could not divide it in two – for example first for connections between tests of impairment assessment and second for connections between biomechanical results of simulator test? You can base your report on more papers, which would be more readable and scientifically more detailed.

2. it is not quite clear from the text if the tests FIST, TIS, SARA, and HDD are used for equestrian qualifications (as it is written in the method part) or are suggested for it (as it looks from the discussion).

3. the paper is huge – the introduction, and discussion covers a lot of info; but statistical analysis seems not comprehensive enough, more description would be better. Perhaps specialists would follow your statistical analysis, the other would find information that a lot of tests were used. Probably at least procedures and/or models would be useful and files (data) that were taken into account by these tests. Some info is given in the discussion part, but it should be in the method part first (see L667).

4. the selection of the control group  - should be completed. It is not quite clear why did you choose the highest level of riders for walk and trot comparison. In my opinion, the suggested dressage test by such level of riders is done by them automatically with almost no attention (results show it also). Especially with automated simulators. Perhaps it would be wise to discuss it in the discussion/limitation part?    How can the change of the control group (lower-level riders) change the results?

5. the mechanics of the simulator and the chosen options should be described earlier in the method part – not only in the discussion.

6. the language is not scientific in some places  – that should be changed as scientific English should be as precise (and simple) as possible – For example L 283 – “ride for scores 10” is not clear for all; L 388 – nine strides, but then three trials? Trial or strides? The trial is not a stride. Did you organize a repetition of trot results? L 593 – scores ranged from floor to ceiling?  L 330 – “nil”? L 636  – developed adults? L852 – ceiling effect is not a well-known idiom

7. citations: the citations are not correct in most cases in the paper –first Animals do not want names with the year as citations, so please exclude it – for example L78, 104,106, 160, and much more.  

8. shortcuts: the paper covers a lot of knowledge and it seems necessary to create the shortcuts list – please remember that shortcuts should be written for the first time – IPC’s – abstract, COM – L 135, IMU- L262; CODA- L345

Detailed remarks:

L 220 – please describe why have you chosen this class of riders and how they were selected.

L 257 – what about the different speeds of living horses?

L 290-291 – was it significant for research? Perhaps attachments should be given and protocols described.

L 292-294 – I do not understand  - because the real dressage test takes 4-5 minutes you have chosen 2 minutes test for comparison? Please explain.

L 330 – what is “nil”?

L342 – six degrees. Why six degrees? How many degrees can be?

L 378 – 0.68 Hz and 0.87 Hz (half of the walk and stride frequency…) please describe these gates according to dressage scaling (corresponding to…)

L 386 – please correct strides/trials being quite clear

L 420-444- as mentioned earlier this part should be more detailed, please be sure that your description would allow for the repetition of the study. Using files/data, models, and procedures should be described as much as possible.

L 429 – could you give the citation why the level of  0.7 not 0.8  was taken?

Table 2, 3, 4 – I cannot see the explanation of (… ) values. sd?

Table 5,6  – please give the meaning of the colors

Table 6 – please give details on lacking data – tables should stay alone –ND? NP?

L 542-550 – repetition of the introduction

L 563-578- does it mean that they are used for the first time in the equestrian sport? That should be underlined in the introduction or as mentioned above (point 1) tests should be used as investigated tests – in the current version they are the key to dividing Para Olympic riders like checked tools. That should be at least clearly written.

L 671/685 – maybe one horse could be used?

L 698 – skill level at the walk? The skill of the horse or the rider?

L 718-719 – six levels of simulated motion – not given in detail in the methods – here text is difficult to follow.

L 784-794 – the last sentence should be changed as it is not true that if there are no significant differences between groups in pelvic symmetry then the COV of R-MAS may be treated as a “specific activity limitation”.

I would suggest ‘specific activity characteristics (not connectable with disability status)” or in other words – specific weight, and height of riders, because this variation is in both groups.  

L 809 –probably horse sensitivity effect could be discussed here as well

L 823 – please describe your point of view wider

L 826 – 783 –the effect of training should be moved/or mentioned once again in the limitation part

L 870 – is not clear, please describe it better

L 890 – it is difficult to follow your point of view, please describe wider/other words. How is it possible that the measured being correlated 0.7 with the others would not be visible in the second left trait?

Conclusions could be more informative and specific.

Author Response

The paper based on the huge project is very interesting, detailed, and professional. Some points should be taken into consideration:

  1. it is a very good report of the project, however, it seems too long for a single paper; the Authors should review if they could not divide it in two – for example first for connections between tests of impairment assessment and second for connections between biomechanical results of simulator test? You can base your report on more papers, which would be more readable and scientifically more detailed.

Thank you for your comments. We have carefully considered your suggestion and the suggestion from another reviewer in reducing the length of the paper. This study was always designed to be a single study and as such we feel it should remain that way. However, we appreciate that we included too much of the background work that went into developing the study within the paper and have removed aspects of the work that can be accessed through other published work. We have also moved some of the results to supplementary as they were not part of the statistical analysis.

  1. it is not quite clear from the text if the tests FIST, TIS, SARA, and HDD are used for equestrian qualifications (as it is written in the method part) or are suggested for it (as it looks from the discussion).

Apologies, we included information about the identification of these tests within the introduction and provided a reference to the scoping review that was conducted to identify them for Para equestrian classification. We have included more information in the introduction and aims to clarify this.

  1. the paper is huge – the introduction, and discussion covers a lot of info; but statistical analysis seems not comprehensive enough, more description would be better. Perhaps specialists would follow your statistical analysis, the other would find information that a lot of tests were used. Probably at least procedures and/or models would be useful and files (data) that were taken into account by these tests. Some info is given in the discussion part, but it should be in the method part first (see L667).

We have included more information related to the statistical tests and linked this back to the aims a little more to clarify what we did statistically and why. We hope this is now clearer.

  1. the selection of the control group - should be completed. It is not quite clear why did you choose the highest level of riders for walk and trot comparison. In my opinion, the suggested dressage test by such level of riders is done by them automatically with almost no attention (results show it also). Especially with automated simulators. Perhaps it would be wise to discuss it in the discussion/limitation part? How can the change of the control group (lower-level riders) change the results?

Both groups were elite level, to compare elite Para athletes with lower level non-disabled dressage rider would not be a true comparison. We conducted pilot testing with a group of 7 lower level non-disabled and 1 lower level Para athlete and can confirm that differences were evident compared to elite athletes. We used walk as our main gait of interest, as walk has previously been identified as a difficult gait to ride, even at elite level, and all athletes (regardless of impairment severity and grade) were able to complete the test at walk, permitting comparisons between participants. This was included in the introduction, but we have now moved it to the methods section.

  1. the mechanics of the simulator and the chosen options should be described earlier in the method part – not only in the discussion.

We have moved details related to the mechanics of the simulator to the methods.

  1. the language is not scientific in some places – that should be changed as scientific English should be as precise (and simple) as possible – For example L 283 – “ride for scores 10” is not clear for all; L 388 – nine strides, but then three trials? Trial or strides? The trial is not a stride. Did you organize a repetition of trot results? L 593 – scores ranged from floor to ceiling? L 330 – “nil”? L 636  – developed adults? L852 – ceiling effect is not a well-known idiom

Regarding ‘ride for scores of 10’: We wished to simulate the competition environment as closely as possible, despite the fact that athletes rode a simulator. To this end, riders were encouraged to ride the test to the best of their ability, thereby simulating the competition environment. By asking all riders to ride as if they were aiming to score ‘10’ for each movement; all riders received the same, standardized verbal instruction prior to testing. We have included the scores used in dressage in Figure 2 for those who are not familiar with the scoring system.

For ‘trials’ apologies, these should all be strides. This has been corrected.

For other comments, we have revised the text.

  1. citations: the citations are not correct in most cases in the paper –first Animals do not want names with the year as citations, so please exclude it – for example L78, 104,106, 160, and much more.

Apologies, this has been corrected.

  1. shortcuts: the paper covers a lot of knowledge and it seems necessary to create the shortcuts list – please remember that shortcuts should be written for the first time – IPC’s – abstract, COM – L 135, IMU- L262; CODA- L345

We have checked shortcuts and provided full descriptions the first time that shortcuts are mentioned. This is except for CODA, as this is not a shortcut, but a name. We have included the company who developed it.

Detailed remarks:

L 220 – please describe why have you chosen this class of riders and how they were selected.

We have included a justification.

L 257 – what about the different speeds of living horses?

We deliberately standardized the speeds of the simulator to ensure that data were comparable between athletes. Studies in living horses have found differences in athlete movement strategies on the same horse or during simulated motion at different speeds.

L 290-291 – was it significant for research? Perhaps attachments should be given and protocols described.

We included the amount of time that the testing protocol took for the performance testing, as it was important that we did not induce fatigue in the Para athletes. We are not sure what additional protocols you would like us to include, as we have described the complete protocol for the performance testing within the methods.

L 292-294 – I do not understand  - because the real dressage test takes 4-5 minutes you have chosen 2 minutes test for comparison? Please explain.

The test was designed to include an appropriate number of individual tasks, in terms of transitions, turns, and periods of riding in a straight line, to enable comparison between the two groups; but without being so long that it may have induced fatigue within some of the Para athletes.

L 330 – what is “nil”?

See previous comments.

L342 – six degrees. Why six degrees? How many degrees can be?

We have included a fuller description. Both the trunk and pelvis were not constrained, so the segments could move in the 3D space in translation or rotation along or about any othrogonal axis.

L 378 – 0.68 Hz and 0.87 Hz (half of the walk and stride frequency…) please describe these gates according to dressage scaling (corresponding to…)

The frequency analysis we used was based on the repeatable pattern of the simulator, which does not fully translate to dressage gait stride frequencies, as the full cycle of the motion pattern is somewhat slower. We have revised and corrected the text.

L 386 – please correct strides/trials being quite clear

Apologies, this has been corrected.

L 420-444- as mentioned earlier this part should be more detailed, please be sure that your description would allow for the repetition of the study. Using files/data, models, and procedures should be described as much as possible.

We have revised this section.

L 429 – could you give the citation why the level of  0.7 not 0.8  was taken?

Apologies, we have re-worded the sentence, as the r values also included strong, but as the level of strong is normally 0.6 to 0.79 we wanted to specify that they were above 0.7. We have removed the descriptors and just included the r value instead.

Table 2, 3, 4 – I cannot see the explanation of (… ) values. sd?

Apologies. These are now included.

Table 5,6  – please give the meaning of the colors

These have been included.

Table 6 – please give details on lacking data – tables should stay alone –ND? NP?

Apologies, this information is now included.

L 542-550 – repetition of the introduction

We have removed this section.

L 563-578- does it mean that they are used for the first time in the equestrian sport? That should be underlined in the introduction or as mentioned above (point 1) tests should be used as investigated tests – in the current version they are the key to dividing Para Olympic riders like checked tools. That should be at least clearly written.

We have revised the aims to clarify this.

L 671/685 – maybe one horse could be used?

Yes, of course this is a possible solution, but the difficulty then is accommodating each Para athletes’ needs to be able to ride the horse (as the simulator provides a safe environment). There are many complications, not least initially helping the athlete to mount the horse that would put the athletes, horse, handlers and researchers at a higher risk. In addition, the welfare of the horse must be taken into consideration, particularly as the practicalities of data collection sessions would mean the horse had to execute multiple tests with multiple unfamiliar riders within a relatively short period of time.

L 698 – skill level at the walk? The skill of the horse or the rider?

Apologies, skill level of the rider. Added athlete

L 718-719 – six levels of simulated motion – not given in detail in the methods – here text is difficult to follow.

Apologies, we have revised this sentence to improve clarity.

L 784-794 – the last sentence should be changed as it is not true that if there are no significant differences between groups in pelvic symmetry then the COV of R-MAS may be treated as a “specific activity limitation”.

I would suggest ‘specific activity characteristics (not connectable with disability status)” or in other words – specific weight, and height of riders, because this variation is in both groups. 

Thank you for your suggestion. We have revised this sentence with your comments in mind.

L 809 –probably horse sensitivity effect could be discussed here as well

This section has been removed completely to reduce the word length, as requested by another reviewer.

L 823 – please describe your point of view wider

This section has been removed completely to reduce the word length, as requested by another reviewer.

L 826 – 783 –the effect of training should be moved/or mentioned once again in the limitation part

We have moved this section to the limitations.

L 870 – is not clear, please describe it better

We have removed this sentence, as this analysis was also removed.

L 890 – it is difficult to follow your point of view, please describe wider/other words. How is it possible that the measured being correlated 0.7 with the others would not be visible in the second left trait?

We have reworded the sentence to improve clarity.

Conclusions could be more informative and specific.

We have revised the conclusions.